# Off-Target Deletion of Conditional *Dbc1* Allele in the *Foxp3*^YFP-Cre^ Mouse Line under Specific Setting

**DOI:** 10.3390/cells8111309

**Published:** 2019-10-24

**Authors:** Chichu Xie, Fangming Zhu, Julie Wang, Weizhou Zhang, Joseph A. Bellanti, Bin Li, David Brand, Nancy Olsen, Song Guo Zheng

**Affiliations:** 1Department of Clinical Immunology, The Third Affiliated Hospital of Sun Yat-Sen University, Guangzhou 510630, China; chichu.xie@outlook.com; 2Shanghai Key Laboratory of Bio-Energy Crops, School of Life Science, Shanghai University, Shanghai 200025, China; fmzhu@ips.ac.cn; 3Department of Internal Medicine, Ohio State University College of Medicine and Wexner Medical Center, Columbus, OH 43210, USA; Julie.Wang@osumc.edu; 4Department of Pathology, Immunology and Laboratory Medicine UF Health Cancer Center, Gainesville, FL 32610, USA; zhangw@ufl.edu; 5Department of Pediatrics and Microbiology-Immunology, Georgetown University Medical Center, Washington, DC 20057, USA; bellantj@georgetown.edu; 6Shanghai Institute of Immunology and Department of Immunology and Microbiology, Shanghai Jiaotong University School of Medicine, Shanghai 200025, China; binli@shsmu.edu.cn; 7Research Service, Memphis VA Medical Center, Memphis, TN 38104, USA; dbrand@uthsc.edu; 8Department of Medicine, Penn State University Hershey College of Medicine, Hershey, PA 17033, USA; nolsen@pennstatehealth.psu.edu

**Keywords:** Foxp3, Cre, Off-target, Germline recombination

## Abstract

The Cre-LoxP conditional knockout strategy has been used extensively to study gene function in a specific cell-type. In this study, the authors tried to engineer mice in which the *Dbc1* gene is conditionally knocked out in Treg cells. Unexpectedly, the conditional *Dbc1* allele was completely deleted with a low frequency in some *Foxp3*^YFP-Cre^ mice harboring floxed *Dbc1* allele under specific settings. It was found that the germline recombination of floxed *Dbc1* allele, which caused *Dbc1* knock out mice, occurred in the male *Foxp3*^YFP-Cre^ mice harboring floxed *Dbc1* allele. Even though the authors documented that *Foxp3* is expressed in the testis, the germline recombination was not caused by the germline expression of Cre, which was driven by the *Foxp3* promoter. The germline recombination may be caused by the unspecific expression of Cre recombinase in the fetus, in which the floxed *Dbc1* allele of some stem cells with development potential to germ cells may be recombined. Additionally, this study found that the floxed *Dbc1* allele was recombined in non-T cells of some *Foxp3*^Cre^
*Dbc1*^fl^ mice, which need to be characterized. Our results also suggest that using male mice with a low frequency of recombined gene allele can reduce the risk of having full knock out mice.

## 1. Introduction

Cre–LoxP technology has been utilized extensively for conditional knockouts in yeast [1], plants [2], cultured mammalian cells [3], T cells [4], neuronal tissue [5] and so forth. Conditional knockout mice using this technology are constructed by breeding a floxed (i.e., flanked by LoxP) gene of interest with mice expressing Cre-recombinase driven by a tissue or cell-type specific promoter. Some examples include the construction of CD4 positive cell knockouts of *Irf4* by breeding *Cd4*^Cre^ mice with those containing floxed *Irf4* sites [6]; breeding dendritic cell driven *CD11c*^Cre^ mice with floxed *TGFRβ*II [7] and the construction of neuronal tissue knockouts of *Rem2* by breeding *Rgs9*^Cre^ with *Rem2*^LoxP/LoxP^ mice [8]. 

Although the Cre-Lox conditional knockout method is an ideal tool for the study of a gene of interest in a specific cell type, some limitations exist [9]. One limitation is the non-specific expression of the targeted gene resulting in off-target recombination in other cell types [10]. Another limitation is the potential for germline recombination of conditional alleles due to the transient Cre expression in the germline [8,9,11,12,13,14]. Thus, a careful characterization of mice generated with this system is an absolute requirement to confirm proper control of conditional recombination. 

Treg cells have been recognized to be crucial in immune tolerance and the prevention of autoimmune responses [15,16]. *Foxp3*^YFP-Cre^ (or *Foxp3*^Cre^) mice were constructed by Rubtsov et al. [17] to study genes related to regulatory T (Treg) cell biology. In the *Foxp3*^YFP-Cre^ mouse, the *Cre* gene is located after an internal ribosome entry site (IRES) sequence which is inserted just after the normal termination codon of *Foxp3*. This mouse was used by several groups to study the function of genes such as *Uqcrsf1* [18], *Id2* [19], *Foxp1* [20], *Maf* [21] in Treg cells. However, Rubtsov et al. [17] and Franckaert et al. [22] have reported the off-target recombination in some hematopoietic lineage cells of a *Foxp3*^YFP-Cre^ mouse. 

The authors have found that the *DBC1* is involved in regulatory T-cell function under conditions of inflammation [23]. To study the specific function of *Dbc1* in regulatory T-cells, *Foxp3*^YFP-Cre^ mice were crossed with floxed *Dbc1* mice to gain the regulatory T-cell knockouts of *Dbc1*. During the breeding process, it was unexpectedly found that the floxed *Dbc1* allele in some mice was totally deleted and the off-target recombination of floxed *Dbc1* allele occurred in tissues other than T cells. 

## 2. Materials and Methods

### 2.1. Mice

All mouse studies were carried out in strict accordance with the guidelines of the Institutional Animal Care and Use Committee at the Sun Yat-sen University. All mice were maintained under specific pathogen-free conditions. The mouse line carrying the *Foxp3*^YFP-Cre^ [17], *CD4*^Cre^ [24] or *Pkm2*^fl/fl^ [25] allele was purchased from the International Mouse Knockout Consortium. The *Dbc1* conditional knockout mice, which contain LoxP sequences flanking exons 4, 5, 6, and 7 were constructed by Shanghai Model Organisms Center, Inc. The breeding strategy is indicated in the text.

### 2.2. Genomic PCR

The genomic DNA (gDNA) was prepared from the toe clips or tail snips of mice. The tissues were lysed by incubation with lysis buffer [100 mM Tris HCl (pH 7.8), 5 mM EDTA, 0.2% SDS, 200 mM NaCl and 100 µg/mL proteinase K] overnight at 56 °C, followed by centrifugation at a speed of 10,000 *g* for five minutes to obtain supernatants with genomic DNA. The DNA was precipitated by isopropanol, washed in 70% ethanol and dissolved in deionized water. The wild type *Dbc1* allele and the floxed *Dbc1* allele were genotyped using the following two primers: 1 (P1, 5’-TCTCACTGTAGCGCAGCCTGAC-3’) and 2 (P2, 5’-ACCCAGCAGAGTACAACAGAAGACAC-3’). The recombined *Dbc1* allele was genotyped by using primer 0 (P0, 5’-CCTGCAGCCCAATTCCGATCA-3’) and primer 2. The wild type *Foxp3* allele and *Foxp3*-IRES-YFP-Cre allele (*Foxp3*^YFP-Cre^ or *Foxp3*^Cre^) were genotyped using the following primer set: (Wt-fwd: 5’-CTATGGAAACCGGGCGATGA-3’, Wt-rev: 5’-AGTGGCAAGTGAGACGTGGG-3’, Cre-fwd: 5’-AGGATGTGAGGGACTACCTCCTGTA-3’, Cre-rev: 5’- TCCTTCACTCTGATTCTGGCAATTT-3’). The PCR product of the wild type *Foxp3* allele is 630 bp, while the *Foxp3*-IRES-YFP-Cre allele is 346 bp. The wild type *Cd4* allele and *Cd4*-Cre allele (*Cd4*^Cre^) were genotyped using the following primer set: (Cd4-Cre-Co: 5’-AACTTGCACAGCTCAGAATGC-3’, Cd4-Cre-Wt: 5’-ACCTGAGATTCCACCAAACTTGA-3’, Cd4-Cre-Mu: 5’-TTAGGGTGGGGCTCAGAAGG-3’) [26]. The PCR product of the wild type *Cd4* allele is 560 bp, while the *Cd4*-Cre allele is 741 bp. The wild type *Pkm2* allele and floxed *Pkm2* allele were genotyped using the following primer set: (P-F: 5’- CCAAAGGATTCCCTTGGGCACAG -3’, P-R: 5’- GCTTTGTCAGAGCTTTGTCACAAATGG-3’). The PCR product of the deleted *Dbc1* fragment was purified using SanPrep Column DNA Gel Extraction Kit (Sangon Biotech Co., Ltd., Shanghai, China) and sequenced by Beijing TsingKe Biotech Co., Ltd in China.

### 2.3. Quantitative PCR (qPCR)

To quantify the floxed or deleted *Dbc1* alleles, the tissues were dissected and gDNA was purified using the Ezup Column Animal Genomic DNA Purification Kit (Sangon Biotech Co., Ltd., Shanghai, China) according to the instructions provided by the manufacturer [27]. The purified DNA was analyzed by real-time PCR analysis with SYBR Green PCR master mix (Takara Biomedical Technology (Beijing) Co., Ltd., Beijing, China). For the floxed *Dbc1* allele, the primer set was Q-mg*Dbc1*-F (5’-TACGAAGTTATTAGGTCCCTC-3’) and Q-mg*Dbc1*-R(5’-CTTGACTTCATCCAAACCG-3’). For the deleted *Dbc1* allele, the primer set was Q -mz*Dbc1*-F (5’-CCAATTCCGATCATATTCAATAAC-3’) and Q-mz*Dbc1*-R (5’-ATAAGCAAGGAAGGTCTGA-3’). The fragment amplified by the primer set of Q-mi*Dbc1*-F (5’-AGAATACTCCACTTTCCCTG-3’) and Q-mi*Dbc1*-R (5’-GTAGACCTCATAAGTGGAGG-3’), which targeted the intron 15 of *Dbc1* served as a normalization control. The quantification of the floxed *Dbc1* allele was normalized against *Dbc1*^fl/+^ mice, while the deleted *Dbc1* allele was normalized against the mouse #4 (*Dbc1*^-/+^). For the *Foxp3* and *Cre* mRNA level, the primer set qm*Foxp3*-F (5’-CCCATCCCCAGGAGTCTTG-3’) and qm*Foxp3*-R (5’-ACCATGACTAGGGGCACTGTA-3’) was used to amplify *Foxp3* [28]. The primer set qmCre-F (5’-CCTTTGAACGCACTGACTTTG-3’) and qmCre-R (5’-GTCCTTCACTCTGATTCTGGC-3’) was used to amplify *Cre*. The fragment amplified by the primer set of qmβActin-F (5’-GGCTGTATTCCCCTCCATCG-3’) and qmβActin-R (5’-CCAGTTGGTAACAATGCCATGT-3’), which targeted *Actb* served as a normalization control. The amplification was quantified using the 2^−ΔΔCT^ method. The qPCR data was plotted as the mean ± standard deviation and analyzed by two-way ANOVA. For all tests, α was set at 0.05.

### 2.4. Immunoblot Analysis

The tissues were dissected and the cells were lysed in RIPA buffer containing 20 mM Tris/HCl, pH 7.5, 1% Nonidet P-40, 0.5% Na-deoxycholate, 135 mM NaCl, 1 mM EDTA, 10% glycerol with 1 mM PMSF, 1 mM Na_3_VO_4_, 1 mM NaF and cocktail protease inhibitor (Sigma, St. Louis, MO, USA). The samples were analyzed by western blotting using the standard procedures [29]. The targeted protein was detected by immunoblot with anti-Foxp3 (14-7979-82; Invitrogen, Carlsbad, CA, USA), anti-GAPDH (#5174; Cell Signaling Technology, Inc., Danvers, MA, USA), and ECL western blotting detection reagents were used (Millipore, Darmstadt, Germany) and visualized using a Tanon 5500 Chemiluminescent Imaging System.

## 3. Results

### 3.1. The Conditional Dbc1 Allele Was Completely Knocked Out in Some Progeny of Foxp3^Cre/Y^ Dbc1^fl/+^ × Foxp3^Cre/Cre^ Dbc1^fl/Fl^ Mice

It was found that *Dbc1* is involved in the maintenance of Treg in an inflammatory environment [23]. To study the function of *Dbc1* in a specific cell type, a transgenic mouse strain carrying a LoxP-flanked (floxed) *Dbc1* allele was generated, in which the LoxP sites were placed upstream of exon 4 and downstream of exon 7 (Figure 1a). In these mice, the wild type or floxed *Dbc1* allele was identified by a PCR assay using primer 1 (P1) and primer 2 (P2) (Figure 1b). The PCR product of the wild type *Dbc1* allele is 384 bp, while the floxed *Dbc1* allele is 488 bp. The mice harboring wild type and floxed *Dbc1* alleles are called *Dbc1*^fl/+^ mice, while the mice containing homozygous floxed *Dbc1* alleles are called *Dbc1*^fl/fl^ mice in this study.

In order to knockout *Dbc1* specifically in Treg cells, the conditional *Dbc1* mice were crossed with *Foxp3*^YFP-Cre^ mice [17], in which the Cre recombinase was under the control of *Foxp3*. In this study, the male mice harboring *Foxp3*^YFP-Cre^ allele are designated *Foxp3*^Cre/Y^ while the female mice are designated as *Foxp3*^Cre/Cre^ or *Foxp3*^Cre/+^. When the female mice homozygous for floxed *Dbc1* and *Foxp3*^Cre^ alleles were crossed with the male mice heterozygous for floxed *Dbc1* and hemizygous for *Foxp3*^Cre^ (*Foxp3*^Cre/Y^
*Dbc1*^fl/+^ × *Foxp3*^Cre/Cre^
*Dbc1*^fl/fl^), it was found that some progeny contained no floxed *Dbc1* alleles, such as the mouse #4 in Figure 2a. This phenomenon could be explained by the fact that the floxed *Dbc1* allele was totally deleted, which could not be detected using primers 1 and 2. To verify this hypothesis, another primer (P0) (Figure 1) was designed to test the deleted *Dbc1* allele. The results showed that the floxed *Dbc1* allele of mouse #4 was deleted in the tail, muscle, brain, liver, immune organs and the reproductive system (Figure 2b). These results demonstrated that the floxed *Dbc1* allele changed to deleted *Dbc1* allele in mouse #4. As the deleted *Dbc1* allele could be also present in the *Dbc1*^fl/fl^ mice (which would not be detected using P1 and P2), the *Dbc1* allele of other mice homozygous for floxed *Dbc1* allele were also analyzed using P0 and P2. The results showed that two mice (#5 and #6) had a deleted *Dbc1* fragment (Figure 2c). Since the deleted *Dbc1* fragment may be from Treg cells, qPCR was used to verify how many floxed *Dbc1* allele were recombined. The results showed that mouse #5 had an equal frequency of deleted *Dbc1* fragment to mouse #4 and floxed *Dbc1* fragment to *Dbc1*^fl/+^ mouse, while mouse #6 was not (Figure 2c). This suggests that one of the floxed *Dbc1* allele in mouse #5 may have changed to the deleted *Dbc1* allele. To further confirm the deleted *Dbc1* allele, the two mice (#4 and #5) were back crossed to wild type mice (Wt). Interestingly, the results confirmed that indeed some of the offspring had deleted *Dbc1* allele (Appendix A). 

To identify whether the deleted *Dbc1* allele was derived from the parent mice, the *Dbc1* alleles of the parent mice were further analyzed. It was observed that one of the mothers also contained a deleted *Dbc1* allele (Appendix A). Therefore, the deleted *Dbc1* alleles in mouse #4 and #5 are derived from the mother 1 mouse. These results suggest that using the Cre/Lox conditional knockout method in the context of *Foxp3* may indeed result in off-target deletion effects.

To verify the deleted *Dbc1* allele, the deleted *Dbc1* fragment (279 bp) was purified and sequenced. The sequence of deleted *Dbc1* fragment is just the sequence of deleted *Dbc1* allele, which supports that the floxed *Dbc1* alleles in mouse #4 or #5 have changed to deleted *Dbc1* alleles (Appendix A).

This study also observed some mice (Appendix A, mouse #P-18 and #P-21) without the floxed *Pkm2* allele in the progeny of male *Foxp3*^Cre/Y^
*Pkm2*^fl/+^ mice crossed with female *Foxp3*^Cre/Cre^
*Pkm2*^fl/fl^ mice, in which the *Pkm2*^fl/fl^ floxed mutant mice were from the Jackson Laboratory [25]. These results suggest that using the Cre/Lox conditional knockout method in the context of *Foxp3* may result in some mice with gene knock out. 

### 3.2. Germline Recombination of Floxed Dbc1 Allele Occurs in the Male Foxp3^Cre/Y^ Dbc1^fl/+^ Mice

The germline recombination of floxed *Dbc1* allele can result in the progeny with a deleted *Dbc1* allele. To verify this and ascertain whether the germline recombination occurred in male mice or female mice, different genotypes of male or female *Foxp3*^Cre^
*Dbc1*^fl^ mice were used to cross with Wt mice. To rule out that the deleted *Dbc1* allele in the progeny was derived from the maternal/paternal mice with deleted *Dbc1* allele, the *Foxp3*^Cre^ mice heterozygous for floxed *Dbc1* allele were chosen to breed (Table 1). 

After identifying the genotypes of progeny mice using primers P1 and P2 or P0 and P2, this study found 1.28% (2 out of 156) mice (#13 and #26) only contained the deleted *Dbc1* allele in the offspring of *Foxp3*^Cre/Y^
*Dbc1*^fl/+^ × Wt mice (Figure 3a, Appendix A). The results of qPCR also were consistent with mouse #13 containing deleted *Dbc1* allele (Figure 3b). It was also observed that a deleted *Dbc1* allele could be found in the offspring of *Foxp3*^Cre/+^
*Dbc1*^+/+^ × *Foxp3*^Cre/Y^
*Dbc1*^fl/+^ mice (Appendix A). In contrast, the genotypes of 119 mice in the offspring of Wt × *Foxp3*^Cre/+^
*Dbc1*^fl/+^ mice were identified, but no mice were obtained with the deleted *Dbc1* allele (Table 1; some results are shown in Figure 3c). Considering that the frequency of the deleted *Dbc1* allele might be lower when mice heterozygous for floxed *Dbc1* allele were used and to rule out that the deleted *Dbc1* allele in the progeny was derived from the maternal mice with deleted *Dbc1* allele, the female *Foxp3*^Cre/Cre^
*Dbc1*^fl/fl^ mice with a lower frequency of deleted *Dbc1* fragment than *Cd4*^Cre/Cre^
*Dbc1*^fl/fl^ mice were also used to cross Wt male mice (Table 1). By genotyping 79 mice in the progeny, not one mouse could be found with deleted *Dbc1* allele (Table 1, some results shown in Appendix A). All the results demonstrated that germline recombination of floxed *Dbc1* allele could occur in the male *Foxp3*^Cre/Y^
*Dbc1*^fl/+^ mice, which could result in progeny with deleted *Dbc1* allele. 

### 3.3. The Floxed Dbc1 Allele Was Recombined in non-T Cells of Some Foxp3^Cre^ Dbc1^fl^ Mice

In the process of genotyping, it was observed that some *Foxp3*^Cre^
*Dbc1*^fl^ mice showed an obvious deleted *Dbc1* fragment, which was amplified by primers P0 and P2, while the others were not (Appendix A). This suggests that the recombination of floxed *Dbc1* allele in these *Foxp3*^Cre^
*Dbc1*^fl^ mice may be various. To check that possibility, qPCR was first used to quantify the recombined *Dbc1* allele in toe and tail, which were tissues used for mice genotyping. The results demonstrated that mice expressing a recombined *Dbc1* fragment obviously as determined by conventional PCR exhibited a greater frequency of recombined *Dbc1* allele (Figure 4a). Additionally, some mice showed a higher frequency of a recombined *Dbc1* fragment than *Cd4*^Cre/+^
*Dbc1*^fl/+^ mice. This indicates that the floxed *Dbc1* allele was recombined in more than T cells in these *Foxp3*^Cre^
*Dbc1*^fl^ mice.

To investigate whether the recombination of floxed *Dbc1* allele occurred in other tissues, two male mice (#B-3 and #G-3) and two female mice (#A-2 and G-1) with a high frequency of deleted *Dbc1* allele in the toe and tail were chosen to measure the extent of *Dbc1* deletion in the brain, muscle, liver, thymus, lymph node, testis, ovary and sperm. As predicted, the *Cd4*^Cre/+^
*Dbc1*^fl/+^ mice showed the highest incidence of *Dbc1* deletion in the thymus and lymph node, while almost no *Dbc1* deletions were detected in other tissues. The *Foxp3*^Cre^
*Dbc1*^fl/+^ mice expressed a detectable but lower incidence of *Dbc1* deletion in the immune organs than *Cd4*^Cre/+^
*Dbc1*^fl/+^ mice. However, the *Foxp3*^Cre^
*Dbc1*^fl/+^ mice with an obvious deleted *Dbc1* fragment in the toe and tail showed many deleted *Dbc1* fragments in the other tissues, while the mice without obvious deleted *Dbc1* did not (Figure 4b). All these results suggest that the floxed *Dbc1* allele is recombined in non-T cells of some *Foxp3*^Cre^
*Dbc1*^fl^ mice. These mice with nonspecific recombination should be ruled out when this kind of conditional knockout mice are used.

### 3.4. Expression of Cre Recombinase in the Fetus Results in Germline Recombination and Dbc1 Knock out Mice

The germline recombination in *Foxp3*^Cre^
*Dbc1*^fl^ mice can be caused by the germline expression of the Cre recombinase driven by a *Foxp3* promoter. Jasurda et al. [30] reported that even though Foxp3 protein was not detected, the *Foxp3* transcript could be found in the testis. In the ENCODE database, the *Foxp3* mRNA was shown to be present in the testis and ovary (Appendix A). The Foxp3 protein in the testis and ovary were also detected using western blotting (Figure 5a). However, this study observed that the two mice (#13 and #26 in Figure 3a, Appendix A and Table 1) with deleted *Dbc1* allele are the progeny of male *Foxp3*^Cre/Y^
*Dbc1*^fl/+^ mice with a high frequency of the recombined *Dbc1* fragment. There were no mice with deleted *Dbc1* allele in the progeny of male *Foxp3*^Cre/Y^
*Dbc1*^fl^ mice with a low frequency of recombined *Dbc1* fragment, even though the floxed *Dbc1* allele is homozygous (Appendix A; Table 1). This study also could not obtain any mice with deleted *Dbc1* allele in the progeny of female *Foxp3*^Cre^
*Dbc1*^fl^ crossed with Wt male mice (Figure 3c and Appendix A; Table 1). This means that the off-target deletion is not dependent on whether the *Dbc1* allele in the breeding mouse is heterozygous or homozygous. Conversely, the off-target deletion depends on whether the breeding mice contain higher frequency of the recombined *Dbc1* fragment or not. The germline recombination probably occurred at the developmental stage of the mice. 

In the EMAGE database (http://www.emouseatlas.org/emap/home.html), Gray et al. [31] and Diez-Roux et al. [32] detected the *Foxp3* expression in the fetus by RNA in situ hybridization. This study also detected the Foxp3 protein, *Foxp3* and *Cre* mRNA in the fetus of *Foxp3*^Cre^ mouse at embryonic day 14.5 (Figure 5b,c). In addition, by quantifying the frequency of the recombined *Dbc1* fragment, it was indeed found that some fetuses of *Foxp3*^Cre^
*Dbc1*^fl/+^ mice (mouse #F3-2, # F3-3, # F3-5 and # F3-7) showed an obvious higher incidence of *Dbc1* deletion than *Cd4*^Cre/+^
*Dbc1*^fl/+^ mice (Figure 5d). These results suggest that the nonspecific expression of Cre in non-T cells of the fetus could activate and excise the floxed *Dbc1* allele. When some cells with deleted *Dbc1* allele develop to germ cells, it can result in *Dbc1* knock out mice in the offspring. Thereby, to limit the risk of having full knock out mice, the male mice with a high frequency of recombined gene allele should be ruled out as breeders. 

Comparing the mice genotype in the progeny of each breeding pair, it seems that the *Foxp3*^Cre^ allele derived from a female breeder causes a higher frequency of the recombined *Dbc1* fragment in the progeny (Figure 3c and Appendix A). Even though the floxed *Dbc1* allele was not inherited from the same female mice harboring the *Foxp3*^Cre^ allele, the floxed *Dbc1* allele still could recombine in some *Foxp3*^Cre^
*Dbc1*^fl^ mice (#106 and #114) (Appendix A). These support that the *Foxp3*^Cre^ allele derived from a female breeder can more easily be expressed and cause off-target recombination in the offspring. 

All the results show that there is a risk of high nonspecific recombination in the *Foxp3*^Cre^ mice. Thus, a careful characterization of *Foxp3*^Cre^ mice with the floxed gene is required.

## 4. Discussion

The data reported here demonstrate that *Dbc1* conditional alleles can be completely deleted when mice are bred with *Foxp3*^Cre^ mice. Although the frequency is not high, it remains possible that the *Dbc1* allele deletion existed in the *Dbc1*^fl/fl^ mice, while the genotype identification using primers flanking only one LoxP site could fail to detect this. Moreover, the deleted *Dbc1* allele is inherited by the offspring. Additionally, this study observed the off-target deletion in the *Foxp3*^Cre^
*Pkm2*^fl^ mice. The off-target deletion may be limited to the conditional *Dbc*1 and *Pkm2* alleles. Otherwise, the off-target deletion may also occur in *Foxp3*^Cre^ mice with other conditional gene alleles. If functional inactivation of the targeted gene requires the deletion of both alleles, it will not affect the results. Nonetheless, when expression of a single copy affects the overall gene function in cells, this off-target deletion needs to be carefully characterized.

It was reported that germline recombination occurred in the *Rgs9*^cre^ line, and that the germline Cre expression could cause this recombination [8]. This study found that *Foxp3* was expressed in the ovary and testis, a finding which is consistent with that of Jasurda et al. [30]. However, the probable Cre expression following the *Foxp3* expression in the testis or ovary of *Foxp3*^cre^ mice did not cause germline recombination. Schmidt-Supprian et al. [33] reported that some floxed genes are more easily be excised than others. Therefore, the *Foxp3* expression in the testis or ovary of *Foxp3*^cre^ mice did not cause germline recombination which might be either as a result of Cre’s unequal sensitivity for conditional gene alleles, or as a result of Cre’s inability to cleave the LoxP site in the final haploid gametes. 

In this report, the fetal Cre expression caused deletion, an interesting finding which to date has not been previously reported. In the fetal development stage, Cre driven by *Foxp3* may function in some embryonic stem cells, resulting in the *Dbc1* allele deletion. These stem cells differentiate into the germline, which resulted in the *Dbc1* deleted offspring.

It is likely that the Cre expression in fetal cells caused the off-target recombination in the fetus of *Foxp3*^Cre^ mice. However, there is a small possibility that the source Cre is from the egg, but less likely from sperm. While the Cre expressed in the egg might not function in the egg itself, it may have functional effects in the early development of the fetus. Although the source of the Cre may indeed have been from the egg, the Cre expression does not occur during the diploid stage of gametogenesis, since all mice with recombined *Dbc1* alleles contain *Foxp3*^Cre^ allele simultaneously. This study detected the *Foxp3* expression in the ovary, but this does not mean that the *Foxp3* expression is in mature eggs. The *Foxp3* expression could not be identified in mature eggs, limited by the number of eggs, which might be resolved by single cell sequencing. 

Even though this study found that *Foxp3* drove fetal Cre function, the frequency of deleted *Dbc1* was variable in the offspring. This means that different mice may contain different levels of the off-target *Dbc1* recombination. Additionally, the off-target recombination possibly occurred in one or both of the two *Dbc1* alleles in non-Treg cells, which results in the same type of non-Treg cells containing different levels of the Dbc1 protein. The variation of the off-target *Dbc1* recombination implies that the fetal Cre expression is transient and the resultant gene recombination is random. The stochastic *Foxp3* expression in the fetus shown by Gray et al. [31] and Diez-Roux et al. [32] also provides support. 

When Rubtsov et al. [17] generated *Foxp3*^Cre^ mice, they used a ROSA26YFP (R26Y) recombination reporter allele in *Foxp3*^Cre^ mice to examine the specificity of Cre-mediated recombination. They observed that *Foxp3*^Cre^ mice exhibited varying degrees (2–10%) of R26Y recombination in different hematopoietic lineage cells. Further, Franckaert et al. [22] observed the deletion of CD28 on conventional CD4^+^ T cells in *Foxp3*^Cre^
*Cd28*^fl/fl^ mice. When considered in the context of our results, the off-target recombination in different hematopoietic lineage cells could indeed be as a result of recombination in the fetus. 

In this report, it was found that the off-target deletion and many somatic cells rather than Treg cells may contain the deleted floxed gene in the *Foxp3*^Cre^ line. The off-target deletion can be genotyped using primers that flank the entire floxed locus of a conditional gene in combination with primers flanking a single LoxP site as suggested by Song et al. [9]. Furthermore, our results suggested that avoiding the use of male mice with a high frequency of recombined floxed allele to breed can limit the risk of having full knock out mice in the *Foxp3*^Cre^ line. However, the deleted gene allele can exist in either of the homologous chromatids of the somatic cells and this cannot be detected by ordinary PCR or qPCR. Our results showed that there is no breeding scheme to limit the risk of having mice with an off-target gene recombination. Therefore, to avoid errors, the mice with a low frequency of deleted gene fragments as detected in ordinary PCR or qPCR should be chosen to do the experiment. 

## Figures and Tables

**Figure 1 cells-08-01309-f001:**
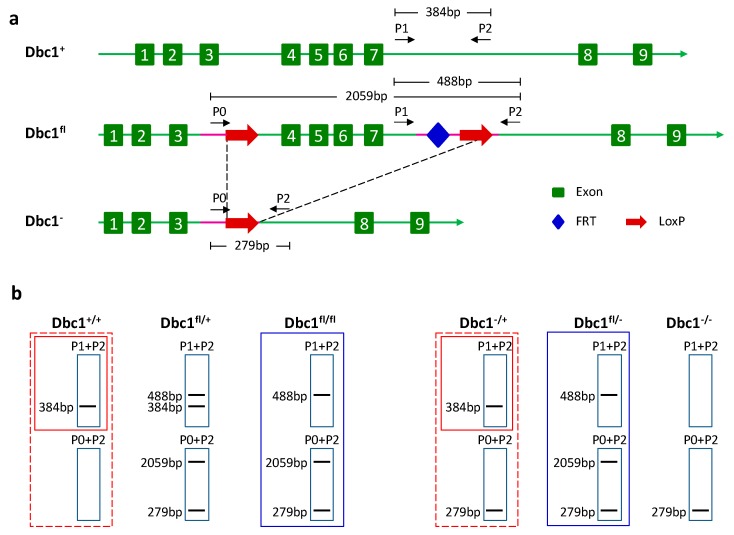
The generation and genotype strategy of the conditional *Dbc1* allele. (**a**) Targeting strategy of the conditional *Dbc1* allele. Shown from top to bottom (i) the *Dbc1* locus with the exons, the site of the primer 1 (P1) and primer 2 (P2); (ii) the targeted locus with the site of the primer 0 (P0), LoxP and FRT sites; (iii) the locus after Cre-mediated deletion of the LoxP sites. (**b**) The genotype strategy of different *Dbc1* alleles. The figure shows the predicted site and number of fragments of the mice with different *Dbc1* alleles using the indicated primer.

**Figure 2 cells-08-01309-f002:**
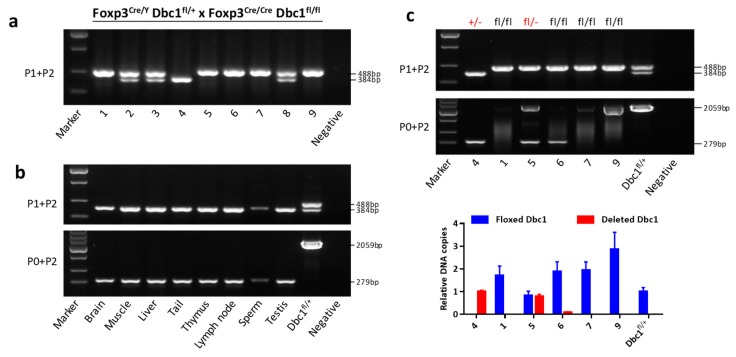
The conditional *Dbc1* allele was complete knockout in some offspring of *Foxp3*^Cre/Y^
*Dbc1*^fl/+^ crossing *Foxp3*^Cre/Cre^
*Dbc1*^fl/fl^ mice. (**a**) The PCR product sizes of the offspring of *Foxp3*^Cre/Y^
*Dbc1*^fl/+^ crossing *Foxp3*^Cre/Cre^
*Dbc1*^fl/fl^ mice using the primer indicated. (**b**) The PCR product sizes of the indicated tissue of mouse #4 in (**a**) using the primer indicated. (**c**) Top panel: the PCR product sizes of the *Foxp3*^Cre^
*Dbc1*^fl/fl^ mice using the primer indicated; bottom panel: the relative amount of the floxed *Dbc1* and the recombined *Dbc1* of the mice in the top panel was tested by qPCR. For qPCR, the results were normalized to the DNA sample from *Dbc1*^fl/+^ mice or mouse #4 in (**a**). The data were represented as the mean ± standard deviation of three independent experiments. +/−, fl/fl, and fl/− show the genotype of *Dbc1* alleles.

**Figure 3 cells-08-01309-f003:**
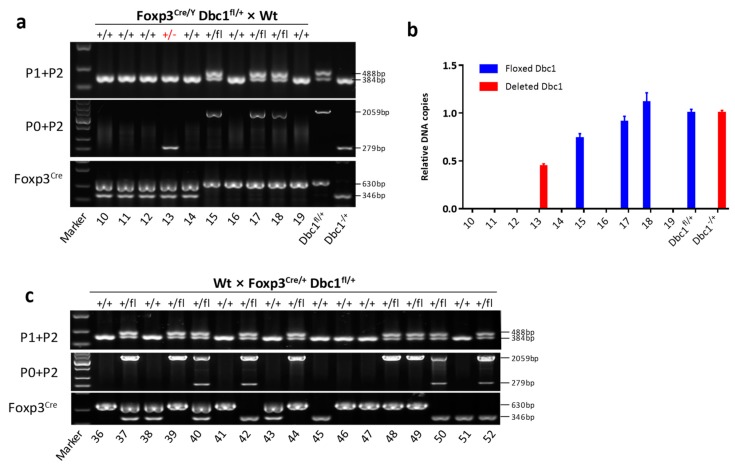
The mice with deleted *Dbc1* allele was present in the progeny of male *Foxp3*^Cre/Y^
*Dbc1*^fl/+^ mice. (**a**) The PCR product sizes of the progeny of male *Foxp3*^Cre/Y^
*Dbc1*^fl/+^ mice crossing Wt female mice using the primer indicated. (**b**) The relative amount of the floxed *Dbc1* and recombined *Dbc1* of the mice in (**a**) was tested by qPCR. For qPCR, the results were normalized to the DNA sample from *Dbc1*^fl/+^ mice or mouse #4 in Figure 2a. The data were represented as the mean ± standard deviation of three independent experiments. (**c**) The PCR product sizes of the progeny of female *Foxp3*^Cre/+^
*Dbc1*^fl/+^ mice crossed with Wt male mice using the primer indicated. The panel of Foxp3^Cre^ shows the genotype of *Foxp3*-IRES-YFP-Cre alleles. +/−, +/+, and +/fl show the genotype of *Dbc1* alleles.

**Figure 4 cells-08-01309-f004:**
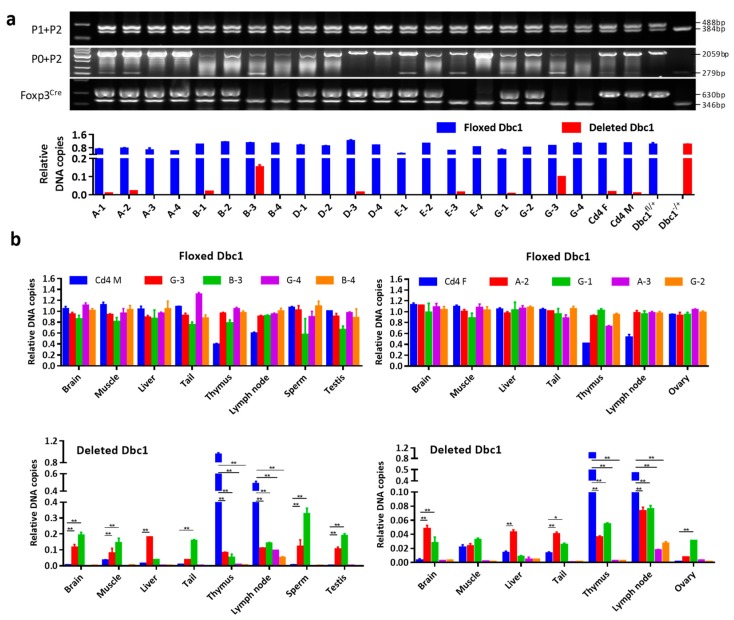
The floxed *Dbc1* allele was recombined in non-T cells of some *Foxp3*^Cre^
*Dbc1*^fl^ mice. (**a**) The relative amount of floxed *Dbc1* and the recombined *Dbc1* of the mice in the progeny of different breeding pairs. The results were compared to the *Cd4*^Cre/+^
*Dbc1*^fl/+^ mice. For qPCR, the results were normalized to the DNA sample from *Dbc1*^fl/+^ mice or mouse #4 in Figure 2a. Mouse #A-1, A-2, A-3, A-4 are offspring of Wt × *Foxp3*^Cre/Y^
*Dbc1*^fl/fl^ mice. Mouse #B-1, B-2, B-3, B-4 are offspring of *Foxp3*^Cre/Cre^
*Dbc1*^fl/fl^ × Wt mice. Mouse #D-1, D-2, D-3, D-4 are offspring of Wt × *Foxp3*^Cre/Y^
*Dbc1*^fl/+^ mice. Mouse #E-1, E-2, E-3, E-4 are offspring of *Foxp3*^Cre/+^
*Dbc1*^fl/+^ × Wt mice. Mouse #G-1, G-2, G-3, G-4 are offspring of *Foxp3*^Cre/+^ × *Dbc1*^fl/+^ mice. The data were represented as the mean ± standard deviation of three independent experiments. The panel of Foxp3^Cre^ shows the genotype of *Foxp3*-IRES-YFP-Cre alleles. (**b**) The relative amount of the floxed *Dbc1* and recombined *Dbc1* of the indicated tissue of the mice in (**a**) was tested by qPCR. For qPCR, the results were normalized to the DNA sample from *Dbc1*^fl/+^ mice or mouse #4 in Figure 2a. The data were represented as the mean ± standard deviation of three independent experiments. “Cd4 F” means female *Cd4*^Cre/+^
*Dbc1*^fl/+^ mouse while “Cd4 M” means male *Cd4*^Cre/+^
*Dbc1*^fl/+^ mouse. * *p* < 0.05; ** *p* < 0.01.

**Figure 5 cells-08-01309-f005:**
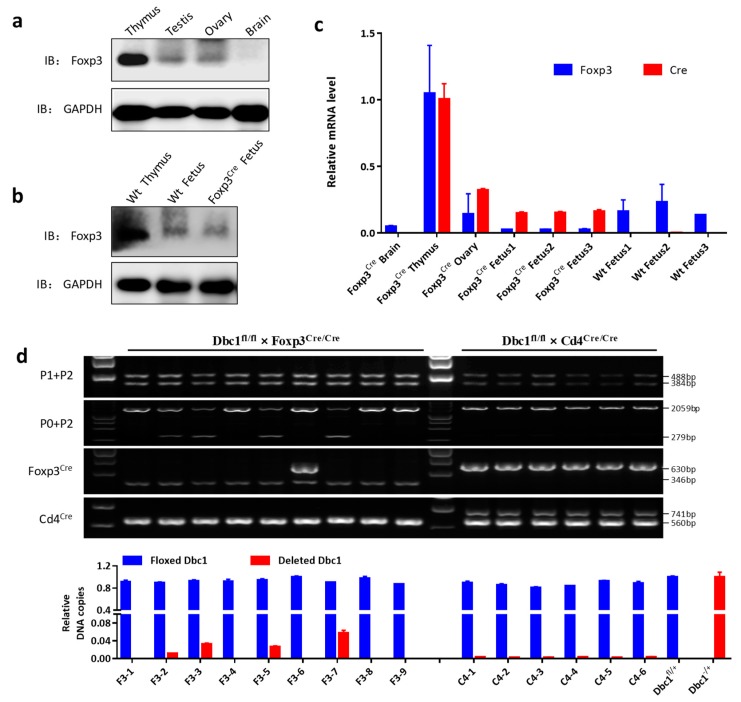
The *Foxp3* was expressed in the ovary, testis and fetus. (**a**,**b**) The Foxp3 protein level in the indicated tissue was tested by western blotting. (**c**) *Foxp3* and *Cre* mRNA level in the indicated tissue was tested by qPCR. The data were represented as the mean ± standard deviation of three independent experiments. (**d**) The PCR product sizes of the fetus at embryonic day 14.5 in the progeny of male *Dbc1*^fl/fl^ mice crossed with female *Foxp3*^Cre/Cre^ mice or *Cd4*^Cre/Cre^ mice, using the primer indicated. The relative amount of the floxed *Dbc1* and recombined *Dbc1* of the fetus was tested by qPCR. For qPCR, the results were normalized to the DNA sample from the *Dbc1*^fl/+^ mice or mouse #4 in (**a**). The panel of Foxp3^Cre^ shows the genotype of *Foxp3*-IRES-YFP-Cre alleles. The panel of Cd4^Cre^ shows the genotype of *Cd4*-Cre alleles. The data were represented as the mean ± standard deviation of three independent experiments.

**Table 1 cells-08-01309-t001:** Frequency of mice with deleted *Dbc1* allele in the progeny of different breeding pairs.

Breeding Mice	Mouse No.	Frequency of Recombined *Dbc1* Fragment	Litter Size	Number of *Dbc1*^−/+^ Mice in Progeny
*Foxp3*^Cre/Y^*Dbc1*^fl/+^ × Wt	YN-3, 4, 5	low	89	0
*Foxp3*^Cre/Y^*Dbc1*^fl/+^ × Wt	YN-1, 2	high	67	2
Wt × *Foxp3*^Cre/+^ *Dbc1*^fl/+^	VN-3, 4	low	55	0
Wt × *Foxp3*^Cre/+^ *Dbc1*^fl/+^	VN-1, 2, 5, 6	high	64	0
*Foxp3*^Cre/Y^*Dbc1*^fl/fl^ × Wt	YM-1, 2	low	75	0
Wt × *Foxp3*^Cre/Cre^ *Dbc1*^fl/fl^	XM-1, 2	low	79	0

“high” means higher frequency of recombined *Dbc1* fragment than *Cd4*^Cre^
*Dbc1*^fl^ mice tested by qPCR; “low” means lower frequency of recombined *Dbc1* fragment than *Cd4*^Cre^
*Dbc1*^fl^ mice tested by qPCR (Appendix A).

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
