# Peer review of "Off-Target Deletion of Conditional Dbc1 Allele in the Foxp3YFP-Cre Mouse Line under Specific Setting"

_cells, 2019, doi:10.3390/cells8111309_

Round 1

Reviewer 1 Report

In manuscript by Xie et. al. author describe off-target deletion of floxed alleles when these mice were bred to Foxp3-Cre mouse. Foxp3-Cre mice are extensively used to study Treg specific deletion and several studies have been reported using these mice. Authors observed very low frequency of deletion (2 mice out of 429 mice screened in various breeding strategies). Although conclusion drawn are appropriate, I worry about the methodology authors have used to test the deletion. PCR and qPCR is routinely used to screen for genetic alteration. However, these methods sometime suffer from technical issues and detection of false positives is not uncommon. Authors have not supported their finding by sequencing the locus to definitively prove loss of allele. Study will also benefit by doing some protein biochemistry or flow cytometry staining to show loss of protein in non-targeted tissues in mice where non-specific deletion has occurred. Below are some comments that authors may find useful.

Authors need to revise title and abstract to reflect key findings that deletion frequency was low and observed in specific settings i.e. when male mice are heterozygous. In addition, authors should include data from homozygous mice bred with WT (or authors should clarify rationale behind using heterozygous mice). Authors should comment on homozygous breeding where both alleles are homozygous for Dbc1 and Foxp3-Cre in both sexes. Do they see more deletion in this setting? It is surprising that only male mice Dbc1 heterozygous bred to WT resulted in germline Dbc1 deletion and no offspring carried deletion when mice were homozygous for Dbc1. One would have expected more mice with germline deletion in homozygous cross, if the proposed hypothesis were correct that egg provided source of Cre. Author need to comment on this. line 205- typo production to be replace with product In total there were 429 mice screened, out of which only 2 mice had off-target deletion, I worry about PCR technical issues. Author need to sequence these mice to prove genetic deletion. Author should also do western blot to prove deletion in non-Treg cells. Line 311, not sure what authors meant by unequal sensitivity. Authors have not shown any data that may conclusively show lack of Cre enzyme activity. Have authors examined pup size from male mice that harbor 10% deletion in sperm cross. Authors reported only 1.2% offspring had deletion. Loss of pups could perhaps explains this.

Reviewer 2 Report

In the manuscript by Xie et al., the authors discovered a recombination defect during the attempt to create a conditional KO mouse using Foxp3Cre-YFP mice to study the role of DBC1 in regulatory T cells. The Foxp3Cre-YFP mouse was created to delete genes in regulatory T cells, based in the assumption that Foxp3 is exclusively expressed in Treg cells. However, the authors reported that, occasionally, the deletion of the floxed allele takes place in other cell types and tissues. Furthermore, deletion of the floxed allele in the germline resulted in the generation of constitutive KO mice.

The present manuscript reveals limitations in the use of the Cre-LoxP system and particularly informs about off-target effects produced in conditional KO mice generated using Foxp3Cre-YFP mice created by Rubtsov et al.

Major concerns:

In line 152 authors wrote “The results showed that two mice (#5 and #6) had a deleted Dbc1 fragment (Figure 2c). Since the deleted Dbc1 fragment may be from Treg cells, we used qPCR to verify how many floxed Dbc1 allele were recombined. The results showed that mouse #5 had deleted Dbc1 allele while the other one had not done so”. The final statement is not clear and the reason for the deletion in mouse #6 is not clearly stated.

In the same experiment, how is compatible the deletion of one floxed allele and, simultaneously, the other floxed allele remains intact in mouse #5?

In line 182 authors wrote “we found 1.28% (2 out of 156) mice (#13 and #26) contained the deleted Dbc1 allele while no floxed Dbc1 allele was present in the offspring of Foxp3Cre/Y Dbc1fl/+ × Wt mice (Figure 3a, Figure S3a)”. However, there are mice in the offspring having a floxed allele (Figure 3: #15, #17, #18).

Based on Figure 3 and Table 1, the authors argue that the recombination takes place in the germline of male mice. Later, in the discussion, they state that “probable Cre expression … in the testis and ovary of Foxp3Cre mice did not cause germline recombination”. First, they have not presented sufficient evidence to conclude that and, second, the messages are contradictory.

How is determined the frequency (low and high) of recombined Dbc1 fragment in Table 1?

In the qPCR experiments, authors indicated that graphs represent data from 3 independent experiments. Since the samples are collected from single specimens, how are the biological replicates generated for 3 independent experiments? Are the errors bars of a single mouse created from technical replicates?

Minor concerns:

Because Foxp3 is encoded in the X chromosome, males are hemizygous for genes encoded the X chromosome instead of homozygous, as stated in line 143.

To refer to the amplicon produced in PCR, the term “PCR product” is typically used, instead of “PCR production”.

In line 182, the authors wrote that they used male or female Foxp3Cre Dbc fl mice. However, in the heading of the fig 3 the mouse genotype indicated is Foxp3Cre Dbc fl/+.

Optimally, unit of centrifugation settings should be provided in g units instead of rpm. Rpm unit is not useful because the force may vary among different centrifuges.

In Fig S3c and d, authors argue that there are differences in the intensity of the 279bp bands among Foxp3Cre Dbc1fl mice and CD4Cre Dbc1fl/+ mice. Since differences in band intensity of independent PCRs may be produced by several factors, quantification of the intensity of the 279bp bands and normalization to equivalent product (e.g.: 2059 bp band of +/fl mouse) would result more convincing.

Round 2

Reviewer 1 Report

Manuscript has much improved after the revision. However, title of the manuscript needs to be changed to mention Dbc1I allele, as author have not done a comprehensive analysis of other floxed alleles that are available or studied/published over the years. Current observation could be limited to Dbcl1 allele, which author needs to acknowledge in their results/discussion.

Author mention in their rebuttal that it was difficult to measure protein because they were unsure whether one or both allele were deleted. Authors need to incorporate this into their discussion, and make their finding easy to understand for the readers. 

I did not find supplementary data provided with this version. 

Author Response

Manuscript has much improved after the revision. However, title of the manuscript needs to be changed to mention Dbc1I allele, as author have not done a comprehensive analysis of other floxed alleles that are available or studied/published over the years. Current observation could be limited to Dbcl1 allele, which author needs to acknowledge in their results/discussion.

Response: We thank reviewer #1 for your supportive comment. Based on your suggestion, the title has been changed to “Off-target deletion of conditional Dbc1 allele in the Foxp3YFP-Cre mouse line under specific setting”. We have modified some sentences in the discussion of this revised manuscript.

Author mention in their rebuttal that it was difficult to measure protein because they were unsure whether one or both allele were deleted. Authors need to incorporate this into their discussion, and make their finding easy to understand for the readers. 

Response: We have modified some sentences in the discussion of this revised manuscript.

I did not find supplementary data provided with this version. 

Response: The supplementary data was uploaded in a separated file in the last version. In this version, the supplementary data was attached behind the main manuscript.

Reviewer 2 Report

I thank the authors for clarifying many of my criticisms. However, it is still not clear to me how was determined the frequency of recombined Dbc1 fragment in the breeding mice indicated in Table 1. In the revised version of the table, the authors indicate that the frequency of recombined Dbc1 fragment is higher or lower compared to CD4Cre Dbc1fl mice tested by qPCR. Could the authors show an example of such test? 

Minor issue: In Figure 5d, the right part of the PCR picture shows the results of the Dbc1fl/fl x Cd4Cre/Cre progeny. However, in the figure, the Cre band is indicated as Foxp3Cre , which is true for the progeny of Dbc1fl/fl x Foxp3Cre/Cre (left side of the picture).

Author Response

I thank the authors for clarifying many of my criticisms. However, it is still not clear to me how was determined the frequency of recombined Dbc1 fragment in the breeding mice indicated in Table 1. In the revised version of the table, the authors indicate that the frequency of recombined Dbc1 fragment is higher or lower compared to CD4Cre Dbc1fl mice tested by qPCR. Could the authors show an example of such test? 

Response: The genotypes and frequency of recombined Dbc1 fragment of the mice chose to breed in Table 1 were added in this revised manuscript (new Figure S4).

Minor issue: In Figure 5d, the right part of the PCR picture shows the results of the Dbc1fl/fl x Cd4Cre/Cre progeny. However, in the figure, the Cre band is indicated as Foxp3Cre , which is true for the progeny of Dbc1fl/fl x Foxp3Cre/Cre (left side of the picture).

Response: In Figure 5d, the right part of the PCR picture shows the results of the Dbc1fl/fl x Cd4Cre/Cre progeny and the left part of the PCR picture represents the results of the Dbc1fl/fl x Foxp3Cre/Cre progeny. The panel of “Foxp3Cre” shows the results of identification of Foxp3Cre allele. As showed in the Figure 5d, the Foxp3 alleles in the progeny of Dbc1fl/fl x Cd4Cre/Cre were wild type Foxp3 alleles. In this revised manuscript, the identification of Cd4Cre allele was added.

This manuscript is a resubmission of an earlier submission. The following is a list of the peer review reports and author responses from that submission.

Round 1

Reviewer 1 Report

Title needs to be restructured/changed as it is not clear what the meaning is in its current form

Materials and methods

For analysis of the qPCR data, the text states quantification was performed using 2-Dct, should this be 2-DDct?

Inconsistencies in M&M and results: No statistics are reported throughout the paper. In the M&M, the authors state a two-way anova was used, but it is not clear which groups they are comparing, and no p-values are reported in the figures. M&M states mean and SEM are shown, figure legends state mean and SD.

Normalisation methods for qPCR should be clarified as it is not clear what method was used exactly. In the results section, results were normalised against a calibrator sample?

Antibodies used for western blot missing

Results

In figure 1, the authors explain the breeding strategy used to generate conditional knock-out mice and the primer sets they use to genotype mice that express a wild-type Dbc1+ allele, a floxed Dbc1fl allele, or a knock-out Dbc1- allele.

The bottom panel in figure1 should be simplified as it is not very clear and arguably makes it harder to understand the cre-Loxp system. Perhaps removing the band sizes and adding a diagram with just the floxed, +/+, and -/- would make it easier to interpret. Instead of showing the primer combinations, it would be easier for the reader if the allele was indicated at the right size (next to the DNA ladder) on the pictures of the agarose gels, i.e. Dbc1+, Dbc1fl, Dbc1-. After explaining the breeding strategy in figure 1, figure 2 shows that some of the Dbc1fl/flFoxp3cre mice show complete deletion of the Dbc1 allele, a finding that has previously been reported by Frankaert et al.  The qPCR data in Figure 2 C is not presented very clearly and the manuscript would benefit from clarifying/revisiting how the qPCR data should be analysed. Sample #4 from figure 2 is used to normalise the qPCR data for all figures, but a better control would be DNA obtained from a knock-out mouse (obtained as such) and not one of the experimental samples.

In figure 4 the authors investigated whether Dbc1 is deleted in the cells of different tissues such as the brain, muscle, liver etc. They observe deletion of Dbc1 is these tissues and conclude that this means that there is non-specific deletion of Dbc1 in non-haematopoietic cells. However, this is a flawed argument as they do not isolate non-haematopoietic cells but whole tissues. Treg cells are known to reside in non-lymphoid peripheral tissues (one example of muscle Tregs : 24315098). It is therefore highly probable that the deletion that is detected, is due to the deletion occurring in Treg cells, or conventional CD4 T cells.

The conclusion that expression of the cre recombinase in the foetus results in germline recombination is not supported by experimental evidence. The data show that Foxp3 can be detected using WB in cells derived from a fetus (there are two bands on the blots though, what is the second band that is cut off?), and that cre can be expressed in the fetus at the RNA level. This might lead to germline recombination but it might also suggest that there are some Foxp3+ expressing Treg cells present in the foetus, even though other reports suggest Treg cells only develop later.

Discussion: I assume the first three sentences of the discussion is a copy/paste of the ‘instructions for authors’.

Nuancing of statements required

Reviewer 2 Report

Xie et al reported an "off-target" deletion of floxed genes by nonspecific expression of Cre recombinase in non-Treg cells in Foxp3-YFP-Cre mice. The authors show even full knockout of Dbc1 gene in some Foxp3-YFP-Cre mice harboring floxed Dbc1 allele. The authors suggest that the unexpected full-knockout of floxed allele in Foxp3-YFP-Cre mice was caused by a transmission of a stochastically-deleted floxed allele in a male germline cell to an offspring. Authors did provide several data which indicates the "promiscuous" expression of Cre recombinase in Foxp3-YFP-Cre mouse, however, as described below, some of the author's conclusions were contradictory to their data, or were overstatement. I also thought some data were at qualitatively and quantitatively unsatisfactory levels. As the Foxp3-YFP-Cre mice have been widely used as a popular tool for Treg cell analysis, publication of this paper is expected to be highly influential to this research field. Thus, I think it is not appropriate to publish this paper as a definitive report in this current premature form.

Major points:

1. Fig.2 disclosed a strong contradiction in the author's interpretation that the full-knockout of floxed allele in Foxp3-YFP-Cre mice were caused by the transmission of stochastically deleted floxed alleles in germline cells of male mice (as emphasized in 3.2 with the subheading "Germline Recombination of Floxed Dbc1 Allele Occurs in the Male Foxp3Cre/Y Dbc1fl/+ Mice"). Because, as for the data shown in Fig. 2, mouse #4 was apparently fl/+ genotype with complete deletion of the floxed allele, however, as the mouse was the offspring of Foxp3Cre/Y Dbc1fl/+ male and Foxp3Cre/Cre Dbc1 fl/fl female, the floxed allele should be provided by the mother mouse. It means, at least in this mouse, complete deletion of the floxed allele must be transmitted from a female germline cell. Authors must deal with this contradiction and reconstruct the manuscript accordingly.

2. qPCR experiment shown in Fig. 2C is not convincing, as the result is inconsistent with the genotyping data shown in the upper panel. Specifically, although #5 shows equivalent strength of the larger (undeleted) and smaller (deleted) bands, and #6 shows mostly with smaller band in the genotyping data, these results were apparently reversed in the qPCR data. In addition, although #1 and #7 were with fl/fl genotype, no or very faint bands can be seen, but instead show the smear bands. Collectively, I strongly doubt the quality of the data shown in Fig. 2c. In addition, it is better to show the sites of primer sets used in the qPCR analysis, with the predicted sizes of the PCR products, just as in Fig. 1a.

3. In 3.4, authors concluded that Cre expression in the testis or ovary of Foxp3-YFP-Cre mice did not cause germline recombination, but instead the Cre expression in fetal cells caused deletion of the floxed alleles, with descriptions "However, we observed that the two mice (#13 and #26) with deleted Dbc1 allele are the progeny of male Foxp3Cre/Y Dbc1fl/+ mice with high frequency of recombined Dbc1 fragment. There were no mice with deleted Dbc1 allele in the progeny of male Foxp3Cre/Y Dbc1fl mice with low frequency of recombined Dbc1 fragment, even though the floxed Dbc1 allele is homozygous (Figure S4b, c; Table 1). We also could not obtain any mice with deleted Dbc1 allele in the progeny of female Foxp3Cre Dbc1fl crossed with wild type male mice (Figure 3c, S3c; Table 1)". However, these observations never officially preclude the possibility that gene recombination occurred in gametes.

Minor points:

1.  Please denote what "Foxp3Cre" panels represent in Figs. 3a, 3c, 4a, and 5d represent, describing what the smaller and larger bands represent.

2. Please provide a detailed description for Foxp3 antibody used in Fig. 5.

3. Please denote what "cd4 F" and "cd4 M" represent, in Fig. 4a.